# Lymphedema and Post-Operative Complications after Sentinel Lymph Node Biopsy versus Lymphadenectomy in Endometrial Carcinomas—A Systematic Review and Meta-Analysis

**DOI:** 10.3390/jcm10010120

**Published:** 2020-12-31

**Authors:** Rianne J.A. Helgers, Bjorn Winkens, Brigitte F.M. Slangen, Henrica M.J. Werner

**Affiliations:** 1Faculty of Health, Medicine and Life Sciences, Maastricht University, 6202 AZ Maastricht, The Netherlands; riannehelgers@gmail.com; 2Department of Methodology & Statistics, CAPHRI, Care and Public Health Research Institute, Maastricht University Medical Centre, 6202 AZ Maastricht, The Netherlands; bjorn.winkens@maastrichtuniversity.nl; 3Department of Obstetrics and Gynecology, GROW, School for Oncology and Developmental Biology, Maastricht University Medical Centre, 6202 AZ Maastricht, The Netherlands; brigitte.slangen@mumc.nl

**Keywords:** endometrial cancer, sentinel lymph node, lymph node dissection, lower-extremity lymphedema, post-operative complications

## Abstract

*Background*: Lymph node dissection (LND) is recommended as staging procedure in presumed low stage endometrial cancer. LND is associated with risk of lower-extremity lymphedema and post-operative complications. The sentinel lymph node (SLN) procedure has been shown to have high diagnostic accuracy, but its effects on complication risk has been little studied. This systematic review compares the risk of lower-extremity lymphedema and post-operative complications in SLN versus LND in patients with endometrial carcinoma. *Methods*: A systematic search was conducted in PubMed and Cochrane Library. *Results*: Seven retrospective and prospective studies (total *n* = 3046 patients) were included. Only three studies reported the odds ratio of lower-extremity lymphedema after SLN compared to LND, which was 0.05 (95% CI 0.01–0.37; *p* = 0.067), 0.07 (95% CI 0.00–1.21; *p* = 0.007) and 0.54 (95% CI 0.37–0.80; *p* = 0.002) in these studies. The pooled odds ratio of any post-operative complications after SLN versus LND was 0.52 (95% CI 0.36-0.73; I^2^ = 48%; *p* < 0.001). For severe post-operative complications the pooled odds ratio was 0.52 (95% CI 0.28–0.96; I^2^ = 0%; *p* = 0.04). *Conclusions*: There are strong indications that SLN results in a lower incidence of lower-extremity lymphedema and less often severe post-operative complications compared to LND. In spite of the paucity and heterogeneity of studies, direction of results was similar in all studies, supporting the aforementioned conclusion. These results support the increasing uptake of SLN procedures in endometrial cancer.

## 1. Introduction

Endometrial cancer (EC) is the second most common gynecologic malignancy and the fourth leading cause of cancer death in women worldwide, with 382,069 new cases and 89,929 deaths in 2018 [1]. EC is mostly diagnosed in postmenopausal women, with up to 20% of women diagnosed before menopause [2,3]. The majority of cases are diagnosed at early-stage due to early symptomatology, mostly abnormal uterine bleeding, in 90% of cases [4]. The primary treatment of endometrial cancer is surgical (total hysterectomy and bilateral salpingo-oophorectomy) to remove and classify the tumor [4,5]. Lymph node dissection (LND) is recommended in clinically presumed low stage disease as a staging procedure according to the International Federation of Gynecology and Obstetrics (FIGO) system [6,7,8]. Lymph node involvement is one of the most important prognostic factors in EC [9,10,11]. However, LND does not improve survival according to two performed randomized trials, and results in increased peri- and post-operative morbidity compared to hysterectomy and bilateral salpingo-oophorectomy alone, including extended operating time and risk of lower-extremity lymphedema [9,12,13,14]. Due to the fact that most patients are diagnosed at an early stage, the overall survival rate of EC is generally favorable ranging between 74% and 91%, and most patients will survive long-term [15]. Factors known to affect EC survival include patient age, tumor subtype and grade, but also comorbidities and post-operative complications [15]. Because of the favorable survival rates, it is important to identify and hopefully modify factors adversely affecting quality of life (QOL) long term, including treatment-related effects. Lower-extremity lymphedema is one of the most cumbersome complications after EC treatment, characterized by chronic and progressive lower-limb swelling, discomfort and dysfunction [16,17,18]. The edema can, in addition to the lower extremities, also manifest itself in the lower abdomen or lumbar region. Incidences of lower-extremity lymphedema fluctuate widely across different studies, with reported ranges between 0% and 37.8% [11,19,20,21,22]. This wide range could be attributed amongst others to underreporting, different methods of diagnosing lower-extremity lymphedema, different rates of additional external beam radiotherapy, and different incidences of comorbidities, such as obesity, affecting its incidence [11,16,17,23].

The sentinel lymph node (SLN) procedure has been explored in EC as an alternative to LND to assess the nodal status [24]. The sentinel node is the first lymph node draining the primary tumor, therefore theoretically reflecting the status of the entire lymphatic basin [25]. Radioactive tracer and/or colored dye are injected to locate the sentinel node, after which it is (micro-) dissected and analyzed. This way, only one (or two) lymph node(s) on either side are removed for staging, thereby potentially decreasing the complications seen after LND [6]. Diagnostic accuracy of SLN to detect metastatic nodes in high risk EC has been reported in literature with proven high sensitivity in a multi-center, prospective cohort study [26]. Also, in a number of retrospective studies, the progression free survival was comparable in similar cohorts independent of SLN or LND [27,28,29,30,31]. Recent years have seen a steep increase in the performance of the procedure. In spite of this, few studies have investigated the effect on associated morbidity/complication rates. Therefore, the aim of this systematic review is to provide an overview of the existing literature on the differences in short- and long-term risks, defined as post-operative complications and lower-extremity lymphedema, after SLN versus LND in EC patients.

## 2. Methods

This systematic review was conducted following the PRISMA statement guidelines [32,33].

### 2.1. Search Strategy

A systematic search was conducted in the PubMed and the Cochrane Library databases by two of the authors (R.J.A.H. and H.M.J.W.) between September 29, 2019 and March 18, 2020 to find studies examining the effect of the SLN versus LND on lower-extremity lymphedema and post-operative complications in EC. In the search, the following terms were used: ‘endometrial neoplasms’, ‘endometrial carcinoma’, ‘endometrial cancer*’, ‘endometrial tumo*’, ‘sentinel lymph node’, ‘sentinel node’, ‘SLN’, ‘sentinel biopsy’, ‘postoperative complications’, ‘post-operative complications’, ‘complications’, ‘side-effects’, ‘side effects’, ‘adverse effects’, ‘lymphedema’, ‘lymph edema’, ‘lymphoedema’, ‘lymph oedema’, ‘edema’, ‘oedema’, ‘lymph node excision’, ‘lymphadenectom*’, ‘LND’ and ‘lymph node dissection’.

Cohort studies and clinical trials were included if they focused on EC, if SLN was compared to LND and if lower-extremity lymphedema and/or post-operative complications were a defined outcome measure. Also, the articles should be written in English, full text should be available and over 10 cases should be reported. Exclusion criteria included case report studies and reviews. Results of the search were screened by two of the authors (R.J.A.H. and H.M.J.W.) based on title and abstract. From the resulting articles, full text was obtained. Decisions to include articles were based on the full text. Reference lists of the included articles were screened for additional articles.

### 2.2. Data Extraction

Data were extracted, including study objectives, inclusion interval, outcome measures, surgical approach, and population characteristics. Effect of existing comorbidities could unfortunately not be further evaluated due to high heterogeneity in study reporting. To assess outcomes, information concerning post-operative complications and lower-extremity lymphedema was extracted per intervention group and imported in Review Manager version 5.3 (Cochrane collaboration) [34]. Lower-extremity lymphedema was assessed using different methods, including the Memorial Sloan Kettering Cancer Center Surgical Secondary Events Grading System (MSKCCSSEGS), the Common Toxicity Criteria Version 3.0 classification by a specialized physiotherapist, and a validated patient-reported survey [24,35,36]. Abdominal edema was only assessed in study III. Therefore it could not be systematically compared in the current review [35]. Different grading systems were also applied to assess post-operative complications, including MSKCCSSEGS (Appendix A), Accordion Severity Classification (ASC) (Appendix A), and Clavien Dindo Classification (CDC) (Appendix A) [24,35,37,38,39,40,41]. To enable comparison and in view of their overlap, the MSKCCSSEGS and ASC were reclassified using the CDC system (Table 1). Grading IVb of the CDC does not have an exact equivalent grade in the MSKCCSSEGS and ASC. Only in study VII postoperative complications were reported in more detail, precluding further comparisons [35].

### 2.3. Methodological Quality of Risk of Bias

Risk of bias was assessed by R.J.A.H., B.F.M.S., and H.M.J.W. using the Newcastle-Ottawa scale (NOS) and overall judgment using the Agency for Healthcare Research and Quality standards [42]. In case of disagreement between the authors, discrepancies were discussed after which agreement was reached. Risk of bias was scored low if the selection domain was scored low risk in minimal three subdomains, if statistical corrections were performed for the effect of confounders, including age, BMI, comorbidities, and adjuvant therapy (comparability domain), and if a minimum of two subdomains were scored low risk. Risk of bias was scored high if the selection domain was scored low risk in maximum one subdomain or if the study did not perform statistical corrections for confounders or if the outcome domain was scored low risk in a maximum of one subdomain.

### 2.4. Statistical Analyses

Review Manager Version 5.3 was used [34]. Heterogeneity was assessed using the I^2^ statistic and visual interpretation of the forest plots. An I^2^ of >75% was considered as considerable heterogeneity, 50–90% represented substantial heterogeneity, and 30–60% moderate heterogeneity [43]. Odds ratios (ORs) and *p* values were calculated to compare the effect of SLN versus LND on lower-extremity lymphedema. A *p* value of ≤0.05 was considered statistically significant. A meta-analysis was conducted on any and severe post-operative complications using forest plots and pooled ORs based on Mantel-Haenszel fixed-effects method, because of the small number of studies and low incidence rates [43].

## 3. Results

### 3.1. Search Results

Through PubMed and Cochrane library searches, 125 potentially relevant articles were identified as detailed in Figure 1. After screening, 106 were excluded because they either did not investigate SLN, LND or complications, were reviews or case reports, or assessed different cancer types. Full-text assessment was performed for the 19 remaining articles; 12 articles were further excluded because of above-mentioned reasons (Figure 1). Seven studies were ultimately included in the review, which are numbered from I to VII (Table 2). The numbering is used to reference.

### 3.2. Included Studies

#### 3.2.1. Study Design

Five retrospective studies, one prospective and one combined retrospective/prospective cohort study were included and coded study I–VII as visualized in Table 1 [24,35,36,37,39,41,44]. Studies I–VI were conducted in tertiary care hospitals, study VII was compiled from register data including multiple hospitals (American College of Surgeons National Surgical Quality Improvement Program). Study and patient characteristics are provided in Table 2. Where needed, authors were contacted for additional information.

#### 3.2.2. Participants

From a total of 5600 patients included in the studies, 2554 patients were excluded as only hysterectomy was performed. 3046 patients were finally included in this systematic review, consisting of all patients who had undergone either SLN or LND in the indicated time period, as further specified in the inclusion criteria. Although patient recruitment intervals varied, all patients were included between 2006 and 2018. Only study IV did not specify the end date of patient recruitment for the SLN cohort [39]. Patients’ median age varied from 61.0 to 79.5 years and differed significantly between the two arms in study II [37]. Median BMI varied from 26.8 kg/m^2^ to 38.1 kg/m^2^. In studies II, IV and VII, BMI was significantly different between the SLN and LND cohort [37,39,41].

#### 3.2.3. Intervention

Short- and long-term risks, defined as post-operative complications and lower-extremity lymphedema respectively, were separately assessed in the current review. Studies I-IV, VI and VII investigated the effect of SLN and LND on post-operative complication risk, whereas studies I, III and V specifically investigated the risk on lower-extremity lymphedema (Table 2). Studies I, II, IV, V and VII also studied the effect of hysterectomy alone, but only the groups treated with SLN or LND were considered for this review [24,36,37,39,41]. The retrospective cohort studies (I, II, IV–VII) did not specify for the extent of lymph node dissection, whereas the prospective cohort study (III) did. For comparability reasons, data of the subgroups of the prospective study were pooled. Information concerning surgery was obtained from the medical records in all seven studies. Surgical approach varied within and between studies and included either robotic-assisted laparoscopy, laparotomy, or regular laparoscopy. In studies II, IV, VI and VII, only laparoscopy was performed (Table 2) [37,39,41,44]. In studies I, III and V, laparoscopy was the main surgical approach, but laparotomy was performed in a small percentage (2–20%) [24,35,36]. Conversion rates only were reported by studies I-III and were all below 2.7%.

Adjuvant therapy was reported by studies III-VI and included brachy-, chemo-, and/or external beam radiotherapy. The number of patients receiving adjuvant therapy varied widely among studies (for example 1–31% for chemoradiotherapy). Tracers used to perform SLN included indocyanine green (*n* = 3), patent blue (*n* = 1) and methylene blue (*n* = 1) [24,35,37,39]. Use of patent blue resulted in a lower success ratio of the SLN (80%) compared to indocyanine green (96%–97.9%) [24,35,39]. The average number of sentinel nodes removed after SLN was 4 per patient. In the LND cohorts, the median number of nodes removed per patient varied from 15–36 [36,39,44,45].

### 3.3. Methodological Quality of Risk of Bias

The risk of bias has been evaluated and is indicated in Table 3.

#### 3.3.1. Selection

In study V, only 49% of the patients that were contacted via email participated in the study [36]. Although power was still sufficient (88%), selection bias could not be ruled out, and was judged high risk of bias for study cohort representativeness. In study VII, patients were included from a large database comprising secondary and tertiary care hospitals [41]. The number of contributing hospitals was not stated. Although the SLN cohort was small (*n* = 144) compared to the LND cohort (*n* = 1089), numbers were still comparable to the other studies. Therefore, the study cohort representativeness was judged low risk of bias.

In all studies, participants for both arms were obtained from the same cohort (Table 3) [24,35,36,37,39,41,44]. Ascertainment of the treatment received was obtained from the hospital records in all studies. Only in study V, absence of study outcome (lower-extremity lymphedema) at study start was specified [36]. Absence of post-operative complications prior to surgery was not reported in any study. Although e.g., preoperative existence of urine tract infections or low hemoglobin levels necessitating post-operative treatment cannot completely be ruled out, their likelihood was considered very low and ‘absence of study outcome (post-operative complications) at study start’ was left unscored in the Risk of Bias table (Table 3).

#### 3.3.2. Comparability

Although all seven studies controlled for age and BMI for both arms, only studies II, V and VII controlled for comorbidities, such as diabetes and hypertension [36,37,41] (Table 3). In study III, patients were matched for BMI and uterine size, but not for significant differences in baseline characteristics [35]. The other six studies applied various methods to correct for imbalanced baseline characteristics.

#### 3.3.3. Outcome

In all but studies IV and VI, a specific tool was applied to assess post-operative complications (Table 3) [24,35,36,37,41]. The follow-up periods for lower-extremity lymphedema and post-operative complications were deemed sufficient in six studies. In study I, lower-extremity lymphedema was assessed within 90 days, which was considered too short for all lower-extremity lymphedema to occur [24]. Due to lack of standardized in-hospital follow-up, adequacy of follow-up was judged high risk of bias in study IV [39].

Based on the above-mentioned observations, an overall judgement has been made regarding risk of bias in the included studies. Overall, study III was judged moderate risk of bias in view of significant differences in baseline characteristics and no evaluation of presence of lymphedema at the start of the study [35]. All other studies were judged low risk of bias.

### 3.4. Outcome Evaluation: Lower-Extremity Lymphedema

Studies I, III and V investigated the risk on lower-extremity lymphedema after SLN and LND [24,35,36]. Lower-extremity lymphedema was measured using a different grading system in all three studies (Table 1). In the SLN cohort, lower-extremity lymphedema incidences in studies I, III and V were 0%, 1.3% and 27.2%. In the LND cohort, the incidences were 10.1%, 18.1% and 40.9%. Incidences in study V were remarkable higher compared to the other two studies. The OR of lower-extremity lymphedema after SLN compared to LND ranged between studies: 0.05 (95% CI 0.006–0.365; *p* = 0.007) in study III, 0.07 (95% CI 0.00–0.21; *p* = 0.067) in study I and 0.54 (95% CI 0.365–0.799; *p* = 0.002) in study V. These results suggest a markedly decreased lower-extremity lymphedema incidence after SLN. In studies III and V, this was statistically significant (*p* = 0.003; *p* = 0.002) [35,36]. Results of all three studies are illustrated in Table 4. A meta-analysis was not performed due to the small number of studies available and substantial heterogeneity (I^2^ = 73%) resulting from the different lower-extremity lymphedema assessments and subsequently reported incidences. Only study V ruled out lower-extremity lymphedema prior to the intervention [36]. Adjuvant radiotherapy was reported in studies III and V. In the LND cohorts, more radiotherapy was applied compared to the SLN cohort [35,36]. The incidence of lower-extremity lymphedema does not seem proportional to the frequency of applied radiation therapy.

Although we were unable to compare edema in other localizations, study III did specify incidence of truncal edema and showed a difference in incidence in line with lower-extremity lymphedema (6/105 LND cohort vs. 1/73 SLN cohort).

### 3.5. Outcome Evaluation: Post-Operative Complications

Studies I-III, IV, VI and VII investigated the effect of SLN and LND on post-operative complications, as visualized in Figure 2. The pooled ORs of any, and of severe post-operative complications were 0.52 (95% CI 0.36–0.73; *p* < 0.001) and 0.52 (95% CI 0.28–0.96; *p* = 0.04) among the included studies respectively. Heterogeneity in the analysis on the presence of post-operative complications is moderate (I^2^ = 48%) and mainly caused by study I [24]. Pooled analysis without this study showed reduced heterogeneity (I^2^ = 0%) and pooled OR remained comparable (0.62; 95% CI 0.43–0.92). Follow-up length was adequate in all six studies and varied from 30 to 42 days. The extent of LND differed between studies: study II included only pelvic LND, studies I, III, IV and VI also added para-aortic LND [24,35,37,39,44]. Study VII did not specify the extent of lymphadenectomy [41].

## 4. Discussion

In this systematic review and meta-analysis, the short- and long-term risks, defined as post-operative complications and lower-extremity lymphedema respectively, after SLN and LND procedures in EC patients were assessed.

Both short- and long-term risks were significantly reduced after SLN compared to LND. The incidence of any as well as severe (grade III–V) post-operative complications was significantly lower after SLN compared to LND. This reduction is also clinically very relevant in view of the elderly, often comorbid, population affected by endometrial cancer. Although grading systems differed in all studies, they could, in a nearly 1:1 manner, be converted into the CDC system, thus ensuring comparability (Table 1, Appendix A). Surgical mode of access was comparable across the study arms in the included studies and thus did not contribute to the lower complication rates after SLN [24,35,36,37,39,41,44]. Decreased incidence of post-operative complications results in several benefits, as decreased bodily stress response and decreased surgical morbidity enable enhanced recovery after surgery (ERAS) and shorter hospital stay [46]. ERAS programs comprise a multimodal and multidisciplinary approach to optimize patient recovery after surgery following specific evidence-based recommendations including preoperative, intraoperative, and post-operative aspects [47]. As demonstrated in the included studies, the SLN results in reduced surgery time and blood loss [24,35,36,37,39,41,44]. Also, procedures more often can be performed minimally invasive, ensuring less occurrence of post-operative ileus and earlier mobilization and discharge. As such, SLN further enhances ERAS adherence.

Besides short-term risks such as post-operative complications, importantly, lower-extremity lymphedema was clearly reduced after SLN compared to LND. The reduction in lower-extremity lymphedema may be even more important, because it can result in reduced QOL and increased morbidity in (former) cancer patients. However, with only three studies focusing on lower-extremity lymphedema and especially with the widely varying methodology used, the size of this effect, although studies are unanimous on the direction, needs to be interpreted with caution.

Risk factors for lower-extremity lymphedema after EC include amongst others (extensive) lymph node dissection, post-operative radiotherapy, and obesity [48,49,50]. As visualized in Table 4, there were differences in how often adjuvant radiotherapy was applied, with a higher frequency in the LND cohort. In study III, the LND cohort received even up to three times more frequently radiation therapy, which may have added to the lower-extremity lymphedema risk on top of the effect of the lymphadenectomy procedure [17,48,49,51,52]. However, the increased incidence of lower-extremity lymphedema does not seem proportional to the differences in applied radiation therapy between SLN and LND cohorts. We therefore argue that the increased incidence in the included studies is mainly explained by the surgical differences as radiation therapy may have contributed to, but cannot explain, the effect observed. It was noted that the extent of LND was not identical between and within studies, neither was the number of nodes removed. It is known that radicality of LND may affect the risk of lower-extremity lymphedema and post-operative complication rates, since para-aortic nodal dissection and removal of large number of nodes (n > 30) results in extended surgery times and increased complexity [9,10,11,49]. As we were unable to correct for this imbalance, LND extent remains a possible confounder, although reflecting regular clinical variation. The low incidence of lower-extremity lymphedema in the SLN groups fits with the assumption that a lower number of nodes removed better preserves lymphatic flow and thus avoids lower-extremity lymphedema [4,6]. Since a causal relationship between number of nodes removed and lower-extremity lymphedema incidence has not been investigated in the included studies, the aforementioned assumption could not be confirmed.

Another possible confounding factor includes BMI. Obesity is known to increase risk of lower-extremity lymphedema independent of surgery due to preferential adipose deposition in the lower extremities and impaired lymphatic transport against gravity in the lower extremities leading to subclinical edema [50]. Also, abnormal lymphatic valves, decreased interstitial fluid transport capacity, lymphatic inflammation, and changes in lymph angiogenesis caused by obesity can affect lymphatic function [50,53]. Although patients’ BMI differed substantially between studies (26.8–38.1 kg/m^2^), it did vary much less between the studies focusing on lower-extremity lymphedema development and especially between study cohorts within studies. Therefore, we feel it is unlikely that (differences in) BMI has affected the results and confounded the effects on lower-extremity lymphedema incidence. Ideally though, lower-extremity lymphedema should be ruled out before the intervention, as was done in study V [36]. However, if this had affected the results, it would have resulted in reduced incidence in lower-extremity lymphedema in this study, rather than the higher incidence that was observed, again suggesting that the observed differences are truly due to the surgical procedures.

Challenges in the current review differ for both outcomes. As depicted earlier, lower-extremity lymphedema was assessed using different methodologies, including grading based on medical records, by a specialized physiotherapist and by means of self-report [24,35,36]. The different methods can result in different incidence reporting, with self-reporting likely resulting in high(er) incidences of lower-extremity lymphedema. These differences were noted in the current review and also reported by others [16,17]. This underscores the need for a standardized grading system for studies comparing lower-extremity lymphedema. Follow-up lengths differed widely across the three studies and was deemed suboptimal in one, as lower-extremity lymphedema may start as late as 18 months or more after surgery and thus some cases may have gone unnoticed [16,54]. As such, also the long follow-up in study V may have contributed to the higher percentages of lower-extremity lymphedema in both cohorts [36]. Regarding lower-extremity lymphedema, the above-mentioned heterogeneity in assessment and follow up lengths should be noted as they likely have contributed to the substantial heterogeneity (I^2^ = 73%), prohibiting meta-analysis. In spite of their different methodologies, all studies do agree that lower-extremity lymphedema occurs significantly less often after SLN compared to LND. Although further studies should be encouraged to test this association on prolonged follow-up, it is unlikely that this effect will disappear.

Considering post-operative complications, moderate study heterogeneity (I^2^ = 48%) was mainly caused by one study I [24]. Differences in baseline characteristics, including adjuvant therapy and number of lymph nodes removed, are unlikely to have significantly contributed as they would have contributed similarly to heterogeneity in the analysis of severe complications, which was not observed. There is however a more pronounced favorable effect of the SLN on ‘all post-operative complications’ compared to the other studies whereas this is much more in line with the other studies when considering ‘severe complications’ only. This may thus explain (part of) the observed heterogeneity. Length of follow-up in all six studies studying post-operative complications was 30–42 days and deemed sufficient to identify all complications [24,35,37,39,41,44]. Finally, the SLN procedure in EC was still in its infancy in the inclusion period of all studies. This implies that results for the SLN groups of all studies may further improve, now that more experience in performing SLN has been gained.

As mentioned previously, diagnostic accuracy of the SLN has been proven in literature both prospectively and retrospectively [26,27,28,29,30,31]. The current review is the first to provide a systematic overview of the available literature on lower-extremity lymphedema and post-operative complications after SLN and LND in EC, with thorough quality assessment. Further foundation can be given to the application of the SLN with reported reduced complications. It is considered unlikely that a large randomized controlled trial will be conducted comparing the complications after SLN and LND in view of the high uptake and positive oncological outcomes of the SLN procedure. Therefore, it is needed to consider the evidence from the available retrospective studies to be the best quality evidence we have.

In conclusion, this systematic review further supports the application of SLN, showing a significant decrease in the known complications associated with the LND procedure, including post-operative complications and lower-extremity lymphedema. However, as short- and long-term complications are not non-existent after SLN; the decision to perform any type of lymph node dissections remains to be taken on clear clinicalgrounds.

## Figures and Tables

**Figure 1 jcm-10-00120-f001:**
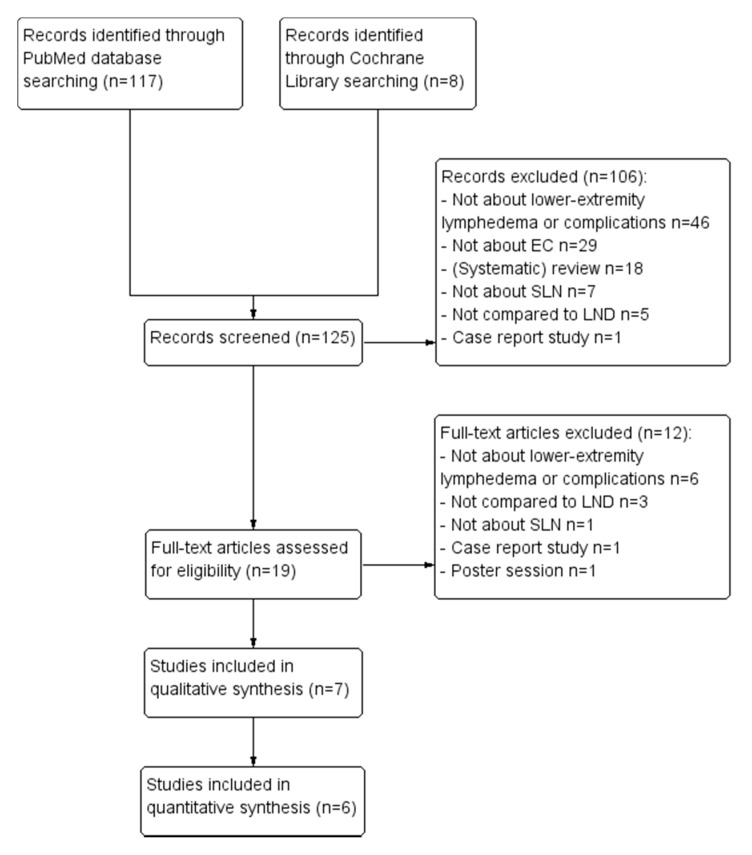
PRISMA flow chart of excluded articles. *Abbreviations*: LND lymph node dissection, SLN sentinel lymph node procedure, EC endometrial carcinoma.

**Figure 2 jcm-10-00120-f002:**
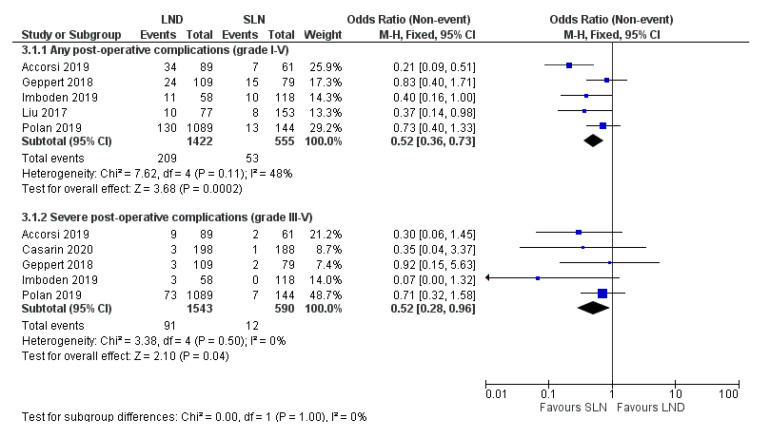
Forest plot on any (3.1.1) and severe post-operative complications (3.1.2) after SLN versus LND. Grading is based on the Clavien Dindo Classification. *Abbreviations*: SLN sentinel lymph node biopsy, LND lymph node dissection. *p* < 0.05.

**Table 1 jcm-10-00120-t001:** Grading systems for post-operative complications.

MSKCCSSEGS	ASC	CDC	
Grade 0	Grade 0	Grade 0	No complications
Grade I	Grade I	Grade I	Bedside care, physiotherapy and oral drugs (antiemetics, antipyretics, analgesics, diuretics, electrolytes)
Grade II	Grade IIa	Grade II	Pharmacological treatment, blood transfusion, parenteral nutrition
Grade III	Grade IIb	Grade III IIIa IIIb	Surgical, endoscopic or radiological intervention No general anesthesia General anesthesia
Grade IV *	Grade III *	Grade IV IVa IVb	Life threatening complications Single organ failure, dialysis Multi organ failure
Grade V	Grade IV	Grade V	Death

*Abbreviations*: MSKCCSSEGS Memorial Sloan Kettering Cancer Center Surgical Secondary Events Grading System, ASC Accordion Severity Classification, CDC Clavien Dindo Classification. * Grading IVb of the CDC has no exact equivalent grade in the MSKCCSSEGS and ASC.

**Table 2 jcm-10-00120-t002:** Descriptive characteristics of the included studies.

Study	Method, Number (n)	Recruitment Setting and Period	Total Follow-Up	Inclusion Criteria	Intervention Groups	Age (yrs) and BMI (kg/m^2^)	Surgical Approach, % Laparoscopy *	Adjuvant Radiation Therapy (%) **	Primary Outcome Measure Considered
**Accorsi (2019) (I)**	Retrospective cohort study (*n* = 250)	Tertiary hospital 2013–2018	90 days	All patients undergoing surgery	HYST + SALP (*n* = 54)	Age 61.0, BMI 31.8	LND 87% SLN 100%	NR	Lymphatic and post-operative complications
					HYST + SALP + SLN (*n* = 61)	Age 60.0, BMI 33.0			
					HYST + SALP + LND (*n* = 89)	Age 62.0, BMI 30.4			
					HYST + SALP + SLN + LND (*n* = 46)	Age 63.0, BMI 29.3			
**Casarin (2020) (II)**	Retrospective cohort study (*n* = 621)	Tertiary hospital 2009–2016	30 days	Patients with early stage EC undergoing robotic surgery	HYST (*n* = 235)	Age 62.5, BMI 37.9	100%	NR	Post-operative complications
					SLN (*n* = 188)	Age 64.1, BMI 35.2			
					LND (*n* = 198)	Age 63.9, BMI 38.1			
**Geppert (2018) (III)**	Prospective cohort study (*n* = 188)	Tertiary hospital 2014–2016	12 months (12–32)	All patients scheduled for robotic surgery	LND + IR + PN (*n* = 85)	Age 68.0, BMI 26.9	LND 97% SLN 98%	LND 31% SLN 10%	Lymphatic and post-operative complications
					LND + IM + PN (*n* = 10)	Age 70.5, BMI 27.7			
					LND + PN (*n* = 14)	Age 73.0, BMI 33.5			
					SLN high risk (*n* = 26)	Age 79.5, BMI 29.7			
					SLN low risk (*n* = 53)	Age 67.5, BMI 28.7			
**Imboden (2019) (IV)**	Retrospective cohort study (*n* = 279)	Two tertiary hospitals *** 2004–2016	33 months	Patients with low risk EC	No LND (*n* = 103)	Age 62.8, BMI 31.0	100%	LND + SLN 17%	Post-operative complications
					SLN (*n* = 118)	Age 62.9, BMI 28.0			
					LND (*n* = 58)	Age 64.8, BMI 29.9			
**Leitao ** **(2018) (V)**	Retrospective cohort study (*n* = 599)	Tertiary hospital 2006–2012	63 months (44-131)	All patients undergoing surgery	HYST (*n* = 67)	Age 61.0, BMI 33.0	80% ***	LND 10% SLN 6%	Lymphatic complications
					SLN (*n* = 180)	Age 61.0, BMI 29.1			
					LND (*n* = 352)	Age 61.0, BMI 29.0			
**Liu (2017) (VI)**	Combined retrospective and prospective cohort study (*n* = 381)	Tertiary hospital 2014–2016	30 days ***	All patients undergoing laparoscopy	Before/LND cohort (*n* = 215)	Age 64.4, BMI 30.5	100%	LND 42% SLN 47%	Post-operative complications
					After/SLN cohort (*n* = 166)	Age 64.5, BMI 31.7			
**Polan (2019) (VII)**	Retrospective cohort study (*n* = 3282)	Secondary and tertiary hospitals 2015–2016	30 days	All patients undergoing laparoscopic surgery	No LND (*n* = 2049)	Age 61.7, BMI 35.8	100%	NR	Post-operative complications
					LND (*n* = 1089)	Age 64.4, BMI 32.7			
					SLN (*n* = 144)	Age 63.0, BMI 36.5			

The studies are numbered I to VII. *Abbreviations*: EC endometrial carcinoma, HYST hysterectomy, SALP salpingo-oophorectomy, SLN sentinel lymph node biopsy, LND lymph node dissection, IR infrarenal paraaortic nodal staging, PN pelvic nodal staging, IM intramesenteric paraaortic nodal staging, NR not reported * laparoscopy includes both robotic and non-robotic approach ** radiation therapy also includes chemoradiation therapy and brachytherapy *** confirmed after personal communication with the corresponding author.

**Table 3 jcm-10-00120-t003:** Risk of bias summary: review authors’ judgements about each risk of bias item for each included study using the Newcastle-Ottawa Scale.

	Representativeness of the SLN Cohort	Selection of the LND Cohort	Ascertainment of Exposure	Demonstration Presence Lower-Extremity Lymphedema at Start of the Study	Controlled for Confounders in Design of Analysis	Assessment of the Outcome	Follow-Up Length	Adequacy of the Follow-Up	Overall Judgement
**Accorsi (2019) (I)**				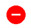			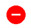		
**Casarin (2020) (II)**									
**Geppert (2018) (III)**				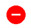	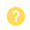				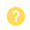
**Imboden (2019) (IV)**								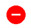	
**Leitao (2018) (V)**	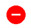								
**Liu (2017) (VI)**						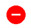			
**Polan (2019) (VII)**									

The studies are numbered I to VII. *Abbreviations*: SLN sentinel lymph node procedure, LND lymph node dissection.

**Table 4 jcm-10-00120-t004:** The effect of sentinel lymph node biopsy and lymph node dissection on lower-extremity lymphedema.

Study	Assessment Method	Intervention Group	Median BMI (kg/m^2^) (Range)	Median nr of Nodes Removed (Range)	Adjuvant Radiation Therapy (%) *	Incidence Lower-Extremity Lymphedema	Odds Ratio	*p* Value **
Accorsi (2019) (I)	MSKCCSSEGS	SLN (*n* = 61)	33 (21.4–48.3)	NR	NR	0 (0.0%)	OR 0.07; 95% CI 0.00–1.21	*p* = 0.067
		LND † (*n* = 89)	30.4 (18.0–46.3)	NR	NR	9 (10.1%)		
Geppert (2018) (III)	CTC Version 3.0 by a specialized physiotherapist	SLN (*n* = 76)	***	5 (0–18)	10%	1 (1.3%)	OR 0.05; 95% CI 0.01–0.37	*p* = 0.007
		LND ‡ (*n* = 83)	***	8 (0–21)	31%	15 (18.1%)		
Leitao (2019) (V)	Lower-extremity lymphedema PRO survey	SLN (*n* = 180)	29.1 (17.9–67.9)	4 (1–21)	6%	49 (27.2%)	OR 0.54; 95% CI 0.37–0.80	*p* = 0.002
		LND † (*n* = 352)	29.0 (18.2–59.1)	19 (1–80)	10%	144 (40.9%)		

The studies are numbered I, III and V. *Abbreviations*: SLN sentinel lymph node biopsy, LND lymph node dissection, MSKCCSSEGS memorial sloan kettering cancer center surgical secondary events grading system, CTC common toxicity criteria, NR not reported, PRO patient-reported outcome. * radiation therapy also includes chemoradiation therapy and brachytherapy ** *p* < 0.05 *** not provided per surgery group † LND included pelvic and/or para-aortic LND ‡ LND included infrarenal and inframesenteric paraaortic and pelvic LND.

## Data Availability

Not applicable.

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
