# Peer review of "Lymphedema and Post-Operative Complications after Sentinel Lymph Node Biopsy versus Lymphadenectomy in Endometrial Carcinomas—A Systematic Review and Meta-Analysis"

_jcm, 2020, doi:10.3390/jcm10010120_

Round 1

Reviewer 1 Report

This article is a systematic review and meta-analysis of the benefits and postoperative complications of sentinel nodes in endometrial cancer.
First, it is a great honor to have been invited to review it for me.
I have then carefully reviewed it.

Major Revise
In the first place, lymphedema can occur in the lower abdomen and lumbar region, as well as in the lower extremities, considering the lymphatic movement of the lymphatic vessels when lymph node dissection is performed in the pelvis. Therefore, it should be mentioned that this is an inadequate assessment of lymphedema. Knowing the travel of the superficial lymphatic vessels, the discussion must be conducted.
Also, it would be difficult to discuss the usefulness of sentinel node dissection from the present data, as it is necessary to interpret the frequency of lymphedema appearance by subtracting the significant effect of radiotherapy from the original appearance.
In addition, postoperative complications were not described in detail, and we believe that more detailed descriptions of postoperative complications are needed.

Author Response

Dear Sir / Madam,

We are very grateful for the constructive comments on the manuscript. We have done our very best to process your suggestions.

Point 1: Lymphedema can occur in the lower abdomen and lumbar region, as well as in the lower extremities, considering the lymphatic movement of the lymphatic vessels when lymph node dissection is performed in the pelvis. Therefore, it should be mentioned that this is an inadequate assessment of lymphedema. Knowing the travel of the superficial lymphatic vessels, the discussion must be conducted.

Response 1: We appreciate your recommendation. We have indeed realized that we did not emphasize that lymphedema can, in addition to the lower extremities, also manifest itself in the lower abdomen or lumbar region. Unfortunately, only one study specify the incidence of truncal edema and showed a difference in incidence in line with lower-extremity lymphedema (6/105 LND cohort versus 1/73 SLN cohort). Since the assessment of truncal edema was only done by one study, these data could not be included in the analysis. Therefore, we have included more information throughout the manuscript, hopefully clarifying the occurrence of lymphedema in other places besides the lower extremities. The adjustments are included in line 57-58, 105-106, 350-352.

Point 2: It would be difficult to discuss the usefulness of sentinel node dissection from the present data, as it is necessary to interpret the frequency of lymphedema appearance by subtracting the significant effect of radiotherapy from the original appearance.

Response 2: Thank you for your remark. We recognize the significant effect of radiotherapy on the incidence of lymphedema. However, it was not possible to distinguish these in the included articles. As a result, the effects could not be evaluated separately. The incidence of lower-extremity lymphedema did not seem proportional to the frequency of applied radiotherapy, as can be seen in the numbers indicated in table 2 and 4. We therefore argue that the increased incidence of lower-extremity lymphedema can mainly be explained by the surgical differences, and that radiation therapy may have contributed to, but cannot explain the effect observed. To further underline this, we made some text adjustments as can be seen in line 348-349 and 400-404.

Point 3: Postoperative complications were not described in detail, and we believe that more detailed descriptions of postoperative complications are needed.

Response 3: Thank you for your recommendation. Post-operative complications were assessed using different classification systems, e.g. the Clavien Dindo Classification. In our belief, this system is a commonly used tool to describe post-operative complications. We understand a more detailed description is preferred. However, this was only done by one study, precluding comparability between studies. Some examples of major complications included cardiac arrest, myocardial infarction, renal failure and sepsis. Minor complications included wound disruption, pneumonia and urinary tract infections. To convey the detailed level of information that one study did provide a detailed overview of the encountered post-operative complications, we added information in line 111-112.

You also commented on some spelling errors that need correction. We carefully re-read the manuscript and revised it for spelling errors and readability. The adjustments can be found in the following line numbers: 38, 40, 47, 108-109, 222, 223, 256, 257, 342, 346-348, 384, 386, 387, 389, table 1, table 2.

We would like to thank you for your considerations and are looking forward to hearing from you.

Yours sincerely,
Rianne Helgers

Also on behalf of,
HMJ (Erica) Werner

Reviewer 2 Report

This study analyzed the postoperative complications and the occurrence of lymphedema in patients affected by endometrial cancer who were treated surgically comparing full lymphadenectomy to sentinel lymph node technique. It is overall well written and well organized. It is an interesting overview of this topic even thoug a mataanalysis could not be performed. The limitations of the study are well exposed in the discussion.

I found some spelling mistakes and I have some suggestions to improve readibility.

I would clarify the process of assignament of risk of bias (paragraph 2.3) and specify why about 2600 patients were excluded from the analysis in paragraph 3.2.2 (I assume they underwent only hysterectomy).

Were the two different cohorts (lymphadenectomy and sentinel lymph node) established on an intention-to-treat? Did any patients in the sentinel lymph node group receive a lymphadenectomy? Another aspect I would expect more details is the surgical route, and data on conversion rate (if available) may be interesting in both groups. I expect open surgery is not represented in the sentinel lymph node group.

Finally I suggest adding a table in which patients' characteristics may be exposed.

Author Response

Dear Sir / Madam,

We are very grateful for the constructive comments on the manuscript. We have done our very best to process your suggestions.

Point 1: I found some spelling mistakes and I have some suggestions to improve readability.

Response 1: Many thanks for critically reviewing the manuscript for spelling errors and readability. In spite of our best efforts, we have not been able to find your suggestions in the review report. We therefore critically reviewed the manuscript ourselves for readability and spelling errors. We made some adjustments in the following lines: 38, 40, 47, 108-109, 222, 223, 256, 257, 342, 346-348, 384, 386, 387, 389, table 1, table 2.

Point 2: I would clarify the process of assignment of risk of bias (paragraph 2.3)

Response 2: Thank you for your remark. We recognize that indeed the process of overall judgement of the included articles was not stated clearly enough in the manuscript. We therefore made some adjustments in paragraph 3 and added information in line 286-288 to further clarify the process. In addition to that, we have altered the arrangement of the text concerning risk of bias and table 3 to improve understanding and readability.

Point 3: I would specify why about 2600 patients were excluded from the analysis in paragraph 3.2.2 (I assume they underwent only hysterectomy)

Response 3: Thank you for your valid remark. We acknowledge that we did not clearly state why about 2600 patients were excluded from analysis. Indeed, in accordance to your assumptions, those patients were excluded as only hysterectomy was performed. We now clarified this in the manuscript in the following lines: 221-222

Point 4: Were the two different cohorts (lymphadenectomy and sentinel lymph node) established on an intention-to-treat? Did any patients in the sentinel lymph node group receive a lymphadenectomy? Another aspect I would expect more details is the surgical route, and data on conversion rate (if available) may be interesting in both groups. I expect open surgery is not represented in the sentinel lymph node group.

Response 4: Thank you for your recommendations. Due to the retrospective nature of most of the included studies, the ‘ intention to treat principle’ was not addressed. Based on your recommendation we critically reviewed the included articles on conversion rates. Only studies I-III reported these rates and were all below 2.7%. We recognize the additional value of adding this information and therefore made an adjustment to the manuscript addressing the conversion rate in the aforementioned studies in line 243-244.

Concerning the surgical route, the majority of the procedures were laparoscopic as indicated in table 2 and 4. Unfortunately, surgical route could not be obtained for the two different arms, even after personal contact with the authors. We therefore included all data we could obtain concerning surgical route in tables 2 and 4.

Point 5: Finally I suggest adding a table in which patients' characteristics may be exposed.

Response 5: Thank you for your suggestion. We regret that we did not present this clearly enough, which may be explained by the fact that most of patients’ characteristics are shown in table 2, however an addition is displayed in table 4. In order to clarify this more, we further elaborated on the patients’ characteristics in the text in line 226-229.

We would like to thank you for your considerations and are looking forward to hearing from you.

Yours sincerely,
Rianne Helgers

Also on behalf of,
HMJ (Erica) Werner

Round 2

Reviewer 2 Report

Thank you for your work on your manuscript. I think it has been improved in readability and overall quality.